# Exploring the Impact of Land Cover on the Occurrence of Ornithobacteriosis and Fowl Cholera: A Case-Case Study

**DOI:** 10.3390/ani15030396

**Published:** 2025-01-30

**Authors:** Lingyu Ouyang, Magnus R. Campler, Sandy Wong, Ningchuan Xiao, Andréia G. Arruda

**Affiliations:** 1Division of Environmental Health Sciences, College of Public Health, The Ohio State University, Columbus, OH 43210, USA; ouyang.193@osu.edu; 2Department of Veterinary Preventive Medicine, College of Veterinary Medicine, The Ohio State University, Columbus, OH 43210, USA; campler.1@osu.edu; 3Department of Geography, College of Arts and Sciences, The Ohio State University, Columbus, OH 43210, USA; wong.484@osu.edu (S.W.); xiao.37@osu.edu (N.X.)

**Keywords:** respiratory poultry disease, poultry health, ornithobacteriosis, fowl cholera, land cover, wetlands

## Abstract

Two major bacterial pathogens, *Ornithobacterium rhinotrachealis* (ORT) and *Pasteurella multocida* (PM), affect commercial turkey farms in the United States (US). This study investigated how different types of land cover, such as wetlands, forests, and urban areas, impact disease occurrences of PM and ORT in 65 commercial turkey farms in the Midwestern US from 2014 to 2021. This study found that farms located closer to wetlands had a higher odds of PM disease compared to ORT. This suggests that turkey farms should be located away from wetlands or implement more stringent biosecurity measures if they are near them.

## 1. Introduction

Respiratory bacterial infections pose a major health challenge for commercial poultry, affecting productivity, animal health, and welfare, and leading to economic losses. Various bacterial pathogens can either be primary or secondary causes of severe infections. Known bacteria that cause respiratory infections for poultry include *Pasteurella* (*P. multocida*, *P. gallinarum*, *P. haemolytica*, and *P. anatipestifer*), *Bordetella* (*B. avium*), *Haemophilus* (*H. paragallinarum*), *Escherichia coli*, and *Ornithobacterium rhinotracheale* (ORT) [1]. The severity of the infection in birds depends not only on the pathogen itself but also on other factors such as management, ventilation, stocking density, temperature and humidity, hygiene, ammonia level, and whether there is a secondary infection [2]. Among all bacterial diseases, fowl cholera and ornithobacteriosis have worldwide occurrence and have been reported in the Midwestern United States (US) [2,3,4,5]. Both of these diseases cause significant economic loss by increasing mortality and decreasing growth [6].

*Pasteurella multocida* (PM) is the causative agent of fowl cholera and has been identified in the US for more than 140 years [7]. *Pasteurella multocida*, considered a primary pathogen [8,9,10], is an important turkey pathogen as outbreaks cause significant economic losses for the commercial turkey industry. Based on an exposure experiment in 1991, overall mortality observed in turkeys exposed to PM was around 51.2% [11]. The signs of PM can be categorized into two stages: acute and chronic. In the acute stage, the major sign is sudden death, with a mortality risk of up to 80% [12]. Birds in the chronic stage of infection show mucous discharges, cyanosis, general depression, ruffled feathers, and diarrhea [13]. Healthy or recovered birds may be lifelong carriers of PM without any symptoms [13]. PM can infect a wide spectrum of animals, including mammals, birds, reptiles, and humans [14,15].

Ornithobacteriosis, caused by ORT, has been recently considered one of the most significant threats in the commercial turkey industry, ranking fourth in the 2020 Turkey Industry Annual Report [16]. *Ornithobacterium rhinotracheale* was first identified in the United States in 1993 and has, since then, become endemic across most countries [16]. The clinical signs of ORT include increased mortality rates and compromised growth in turkeys [6]. The severity of the disease varies depending on environmental factors and synergism with other pathogens [2,6]. The role of ORT as either a primary or secondary pathogen has been debated but seems to be dependent on specific pathogen-strain virulence and residing co-infections. For example, Pan et al. found that ORT in Chinese domestic poultry can induce high mortality alone, while co-infection with H9N2 avian influenza viruses would lead to greater mortality [17]. Experimental inoculation of ORT conducted by Morales-Erasto et al. in Peru showed minimal or no clinical sign of infection when poultry are exposed to ORT alone but severe signs of infection when co-infected with *Avibacterium paragallinarum* [18].

The modes of transmission of PM and ORT share similarities. Both pathogens can be transmitted through aerosol transmission, direct contact with infected birds, or contaminated fomites, such as equipment and facilities [2,19,20,21,22,23,24,25,26]. *Pasteurella multocida* can also be transmitted through contact with infected domestic animals, such as cats, and even insects that had direct contact with infected birds [19,20,21,22,23,25]. In contrast, ORT is able to be transmitted vertically, from parent to offspring through infected eggs, in addition to horizontal transmission [2,24,26]. In commercial turkey industry for the Midwestern US, the “all-in/all-out” strategy minimized the possibility of vertical transmission as only chicks from certified healthy breeders were introduced to the farm.

Land cover refers to the physical material at the surface of the Earth, which includes natural elements like vegetation, water bodies, and soil, as well as human-made structures such as buildings and roads [27]. It is a critical component for environmental monitoring and management, mostly studied through remote sensing techniques to understand and map the distribution of various land cover types [28]. Changes in land use and land cover can significantly impact the dynamics of various infectious diseases by altering the vector, host, and pathogen niches. A change in the host and vector community composition may alter behavior, movement, and spatial distribution of hosts and/or vectors, as well as influence socioeconomic factors, resulting in an increase of environmental contamination [29]. However, past literature has presented limited evidence and discussion about the relationship between commercial poultry respiratory diseases with surrounding land cover. Blanchong et al. investigated whether wetlands could serve as long-term reservoirs for PM, and their results indicated that the pathogen could persist for up to 7 weeks, suggesting that wetlands are unlikely to be long-term reservoirs [30]. Thus, a more in-depth discussion regarding land cover and poultry respiratory diseases is needed. The goal of this study was to assess the importance of land cover as a risk factor for the two above-mentioned poultry bacterial respiratory pathogens, PM and ORT, in commercial turkey farms.

## 2. Materials and Methods

### 2.1. Study Design

This study followed a case-case study design, which is a type of case-control study used to contrast the exposure of study subjects with diseases having related etiological definitions [31]. Both PM and ORT share an etiological definition as they have similar transmission modes, including direct contact, fomite, and aerosols. This study design has been used in epidemiological studies to compare the relative importance of risk factors in different diseases [32]. The primary advantage of a case-case study is its ability to minimize selection and information biases, as both groups of cases share similar clinical signs and were identified within the same production company [31,32].

### 2.2. Source Population and Output Definition

Data used in this study consisted of PM and ORT isolates detected in Midwestern US commercial turkey sites with PM and ORT—related disease occurrences confirmed by bacterial isolation between 2014 and 2021. All sites belonged to one production company that followed the same biosecurity protocols and had the same animal density standards, veterinarian oversight, and overall site design. The sites also adhered to a “one-age” principle, with turkeys entering the farm at 5 weeks of age and exiting at 21 weeks. As the dataset was being created, on-site veterinarians evaluated the symptoms and collected samples based on the severity of the clinical signs (samples were collected 50–80% of the time). After bacterial isolation was conducted by personnel at the production system-level, positive samples were sent to the Animal Disease Diagnostic Laboratory (ADDL) under the Ohio Department of Agriculture (ODA) to be banked and frozen. For this and an accompanying project [33], confirmation of positive farms was conducted at the ADDL, where samples were cultured and molecularly characterized as described below. These processes were conducted separately: in brief, for PM, bacterial cultures were conducted using blood plates incubated at 37 °C for 48 h. Isolate characterization was performed using matrix-assisted laser desorption ionization time-of-flight mass spectrometry (MALDI-TOF MS). Subsequently, PM isolates underwent whole-genome sequencing (WGS) using Illumina MiSeq (San Diego, CA, USA) platform with genome libraries prepared via Nextera XT DNA sample prep kit, with this work being conducted at the University of Maryland. For ORT, bacterial culture was performed in blood agar under microaerophilic conditions at 37 °C for 48 h. Isolates were then confirmed using real-time PCR, and WGS was then performed in a consistent manner as described for PM. Further details on the molecular characterization and genotyping methods are available in Campler et al. [33].

The data analysis for this study was conducted at the farm level and, for turkey sites that recorded multiple disease occurrences (of either ORT or PM), only the first disease occurrence was included. This criterion was implemented due to the fact that multiple entries from the same farm may not be independent from each other. Additionally, farms with positive samples for both PM and ORT (*n* = 6) were excluded from the analysis because these two case groups were exclusive of each other. Finally, only farms with a clearly identifiable barn structure, as observed on the 2022 Google Earth Pro satellite images, were included. This approach was used to confirm that the latitude and longitude were correctly recorded. After selection, a total of 65 farms were included, consisting of 28 PM-positive farms and 37 ORT-positive farms. All sites were located in an area of approximately 150 × 150 km^2^ (Figure 1). Since the study area was relatively small, all the farms were considered in the same geographical region sharing the same climate and weather conditions.

### 2.3. Data Collection for Variables of Interest

Factors hypothesized to influence the odds of ORT- and PM-related disease occurrences included number of poultry farms within 4.5 km, distance to the closest land cover, and season and year of the disease occurrence. The rationale behind the 4.5 km radius threshold was based on findings by Campler et al. [33] who reported a significant spatio-temporal cluster diameter for PM of 9 km for Midwestern commercial poultry farms. Thus, an area with a 4.5 km radius was built to calculate the number of poultry farms around the farm of interest, as this radius represented the plausible non-fomite PM transmission radius. To calculate the number of poultry farms, an object-based image analysis methodology was used [34]. This methodology was adopted to map all poultry farms within the study region. A buffer zone of 4.5 km was created around each PM-/ORT-positive farm using ArcGIS Pro (version 3.2.0). Subsequently, the total number of poultry farms within each buffer zone was calculated. Poultry-farm density calculation was performed in ArcGIS Pro.

The land cover raster layer was acquired from the NLCD 2016 dataset. This is a well-tested publicly available dataset released in 2016, which is in close range to our study period. The available land cover categories in the region defined by NLCD 2016 were open water, four degrees of developed/urban land (open space; low, medium, high intensity), barren land, three types of forest (deciduous, evergreen, mixed), shrub/scrub, grassland, pasture, cultivated crops, and two types of wetlands (woody, emergent herbaceous). The four degrees of urban land cover and three types of forest were aggregated. These land cover types were then vectorized into polygons. Using ArcGIS Pro, the distance from each farm of interest to the nearest land cover was calculated for every category.

The years of outbreak were categorized into 2014–2017, 2018–2019, and 2020–2021 for modeling purposes, with the goal of each category having 20 farms to avoid empty categories. The season of the outbreak was categorized based on astronomical seasons (21 June, 22 September, 21 December, 20 March) using disease occurrence day.

### 2.4. Statistical Analysis

Logistic regression models were built using R (v. 4.4.0) in RStudio (v. 2024.4) to investigate the association between risk factors of interest and the outcomes, namely ORT- and PM-related disease occurrences [35]. Predictor variables included the number of poultry farms within a 4.5 km radius, the year of the outbreak, the season, and the distance to various land cover types (Figure 2) [36].

The model building process began by checking key assumptions of logistic regression, including independence between observations, and linearity (linear relationship between all continuous predictors and the log odds of the outcome). To check for linearity, loess curves were produced [37]. If the linearity assumption was not met, continuous variables were categorized based on their median values [38]. Distance to the nearest pasture was categorized accordingly by median (3956.8 m). The categories were far (≥3956.8 m) and near (<3956.8 m). Two variables, distance to nearest cultivated cropland and distance to nearest urban land, were excluded from the analysis due to insufficient variability. Distance to nearest cultivated cropland had a range of [0, 8.18] meters and 51.5% of the values recorded as 0. Distance to nearest urban land had a range of [0, 498.40] meters and 41.54% of the values recorded as 0. Next, correlations among predictors were assessed using the Spearman correlation coefficient with a cutoff value of 0.80 to identify highly correlated variables. Univariable logistic models were then constructed, employing a conservative *p*-value threshold of 0.2 for screening variables to be included in the full model [39].

Finally, the final model was developed using a backward stepwise method, declaring statistical significance at a *p*-value less than 0.05 [40]. Confounders were initially identified using a causal diagram (Figure 2) and were re-evaluated each time a variable was removed. Variables including the number of poultry farms within a 4.5 km radius, the year of the outbreak, and the season of the outbreak were considered as theoretical confounders. Confounding variables remained in the model if removing them resulted in a change of 20% or more in the odds ratio of the other variables. Additionally, interactions between variables that remained in the final model were explored. The final multivariable logistic regression model fit was assessed using the Hosmer–Lemeshow goodness-of-fit test [41].

## 3. Results

### 3.1. Descriptive Statistics and Univariable Analysis

Descriptive statistics for farms with positive PM and ORT, cross-tabulated by various risk factors, are presented in Table 1. In the univariable analysis, distance to the nearest wetland, forest, and barren land were chosen among all land cover types, along with the season and year of the outbreak as independent variables.

### 3.2. Multivariable Logistic Regression Analysis

The results of the multivariable logistic model are presented in Table 2. Conceptual confounders were evaluated and did not change the odds ratios of other variables by 20% or more; thus, some of them were removed during the stepwise variable selection process. The Hosmer–Lemeshow test showed a good fit (*p*-value = 0.972).

Among all the land cover variables, the only one retained in the final multivariable model was distance to nearest wetland (Table 2). For every 1 m increase in distance from a farm to the nearest wetland, the odds ratio was 0.9976 (*p*-value = 0.004). This represented a decrease of approximately 0.24% in the odds of a PM occurrence compared to an ORT occurrence for each meter increase. More significantly, for every 100 m increase in distance from a farm to the nearest wetland, there was a 20.2% decrease in the odds of a PM disease occurrence compared to the odds of an ORT disease occurrence.

The year of the outbreak also showed significance in the final model. The odds of a PM occurrence in 2018–2019 was 1.48% (*p*-value = 0.001) of the odds in the reference period (2014–2017), which represents a significant decrease. The odds of a PM disease occurrence in 2020–2021 was 6.77% (*p*-value = 0.012) of the odds in the reference period (2014–2017), also indicating a significant decrease compared to the reference period.

## 4. Discussion

To our knowledge, this study is the first attempt to explore the potential of land cover attributes as risk factors for turkey disease occurrence by bacterial pathogens ORT and PM. Every meter closer to wetland increased 0.24% in the odds of PM occurrence compared to ORT occurrence. This study highlights the importance of utilizing commonly collected but often underutilized data in retrospective research. This data can provide insights in disease control and prevention, informing public health strategies without the need for costly new data-collection efforts.

Wetlands having a more significant impact on PM occurrence compared to ORT could be explained by multiple hypotheses. First, the main route of transmission in fowl cholera is related to spillover events from waterfowl or wild birds in wetlands. Wild birds are known as reservoir hosts for PM [9,22,42]. The pathogen could be introduced to the farm through aerosols or direct contact with wild birds, or it could be carried by other farm animals such as cats or rodents. The spillover, and potential spillback of PM between backyard flocks and wild waterfowl, cormorants, and shorebirds, was documented by a study in Denmark in 1998 [43]. In contrast, the main route of ORT transmission may be less associated with wild birds. In this case, even though wetlands are not considered as long-term reservoirs based on their ability to preserve PM, their complex role in disease transmission needs reconsideration [30]. It is possible that ORT may be introduced by wild birds using the same vectors as PM but with the birds remaining asymptomatic until other risk factors are introduced. Future studies are needed to explore wild bird transmission routes further.

While distance to the nearest wetlands was found to be the only significant land cover risk factor in our final statistical model, this does not rule out the possibility that other land cover types have influence on disease dynamics. For instance, the distance to the nearest forest showed significance in the univariable analysis. This finding aligns with previous research, where vegetation cover was found to have the most explanatory power in a study on fowl cholera outbreaks in Common Eiders in the Canadian Arctic [44]. That study suggested that vegetation cover may act as a surrogate for factors such as ground moisture retention and temperature regulation, both of which influence the survival of environmentally transmitted pathogens like PM [45,46,47]. In contrast, barren land, another land cover type that showed significance in the univariable analysis, has limited evidence supporting its role in PM transmission. Barren land refers to areas with no vegetation cover. In our study region, it is the least common land cover type, typically associated with strip-mined areas, limestone quarries, and abandoned coal mines [48,49,50]. One speculation is that the distance to barren land may serve as a proxy for regional development, with closer proximity to barren land indicating less developed areas, which could potentially impact the disease dynamics.

The results of this study can be used for building a more sustainable disease prevention plan for fowl cholera. According to the Avian Disease Manual, the recommended methods of prevention were applying vaccinations and high biosecurity protocols including prompt disposal of carcasses, minimizing contact with other animals, and thorough disinfection of equipment following an outbreak [51]. Building upon the manual, this study could provide guidelines for selecting new farm locations to mitigate the risk of fowl cholera. Additionally, for farms located near wetlands, implementing mandatory vaccinations and establishing a comprehensive fowl cholera surveillance program should be recommended.

Outbreak year was another significant risk factor in the multivariable logistic model. PM incidence was significantly higher in 2014–2017 and lower in 2018–2019 and 2019–2020 compared to ORT. The results demonstrated a higher ORT incidence in more recent years. However, the year of the outbreak could be influenced by unmeasured risk factors, such as changes in on-site veterinarians or variations in the sampling and banking protocols across different years. By incorporating the year, the analysis accounts for these underlying effects that may otherwise go undetected. Additionally, land cover changes over time. While this study does not specifically investigate land cover change as a risk factor, future research could explore this aspect.

In this study, season of the outbreak did not remain in the multivariable logistic model. However, the records indicated that PM outbreaks were most frequent in spring and autumn, while ORT outbreaks predominantly occurred in the spring. Previous research has documented that fowl cholera is more prevalent in late summer, fall, and winter [6]. Evidence suggests that ORT may be influenced by seasonal factors, although there is no consensus on which season exhibits the highest incidence [24]. This study revealed no significant seasonal differences in the patterns of PM and ORT outbreaks within the study region.

Another important piece of information to note when interpreting study findings was that, for our study, we did not have data on “negative” or “healthy” farms, and as such we were unable to use a traditional “case-control” study design. The two main reasons for choosing a case-case study design were data availability and the challenges in defining disease-negative farms because of the lack of purposefully collected and tested samples from diagnostics. First, as a retrospective observational study using secondary data, we used the data that were already collected by the production company. We did not have access to all the farms of the company, but solely those that had a disease incidence over the observed period. During the data-collection process over the years, on-site veterinarians assessed disease occurrences and collected samples based on clinical signs. Farms with positive samples were confirmed to be positive using a combination of in-house bacterial culture and diagnostic laboratory molecular testing. However, for farms without sample collection, we could not definitively confirm they are free of the disease. This lack of certainty prevented us from establishing a reliable pool of “negative” farms for comparison.

This study had several limitations. First and foremost, this research did not directly establish a relationship between PM and wetlands due to the limitations of the case-case study design [31]. Additionally, the sample size was relatively small, as it was a retrospective observational study utilizing data collected from a single production company over the period from 2014 to 2021. Due to the nature of the data collection, it was not feasible to increase the sample size. Future studies with larger sample sizes would be beneficial to validate and expand upon these findings.

We did not incorporate the number of houses/barns on the farm as a risk factor. This decision was based on the assumption that all buildings within a single farm are exposed to the same land cover risk.

Furthermore, datasets used in this study had unmeasured inaccuracy. The NLCD 2016 dataset achieved an overall accuracy of 86.4% for Level II classification and 90.6% for Level I classification [52]. Among all classes of land cover, wetlands had the lowest user accuracy at 68%, which means that out of all pixels classified as wetland, 68% truly represented wetland areas on the ground [52]. The 96% producer’s accuracy value represented 96% of the actual wetland areas in the landscape being correctly classified as wetlands [52]. In other words, in NLCD 2016 [52], the wetland category inaccurately included land cover other than true wetlands, while it accurately recorded almost all wetlands. Forest was the land cover type that was most misclassified as wetland in NLCD 2016. This uncertainty could lead to an overestimation of the risks associated with wetlands, attributing forest-related risk factors incorrectly to wetlands. However, no classification method is perfect, especially for wetlands, due to their seasonal and temporal variability. Despite these limitations, the NLCD 2016 dataset is widely validated and extensively used in ecological and land cover studies. As a low-resolution, publicly available dataset, it was the most accessible and appropriate choice for this study. While the poultry location dataset created under the methodology published by Maroney et al. showed high accuracy in North Carolina and Arkansas counties, its performance in Ohio and Indiana has not been assessed [34].

Last but not least, this study was conducted under the assumption that land cover did not change significantly during 2014–2021. Future research focused on land use and land cover change would be valuable to explore further.

## 5. Conclusions

In conclusion, this study revealed wetlands were associated with an increased odds ratio of PM compared to ORT disease occurrences in Midwestern US commercial poultry farms between 2014 and 2021. These findings can be used to guide the selection of locations for new poultry farms and to develop more stringent biosecurity guidelines for existing farms situated near wetlands.

## Figures and Tables

**Figure 1 animals-15-00396-f001:**
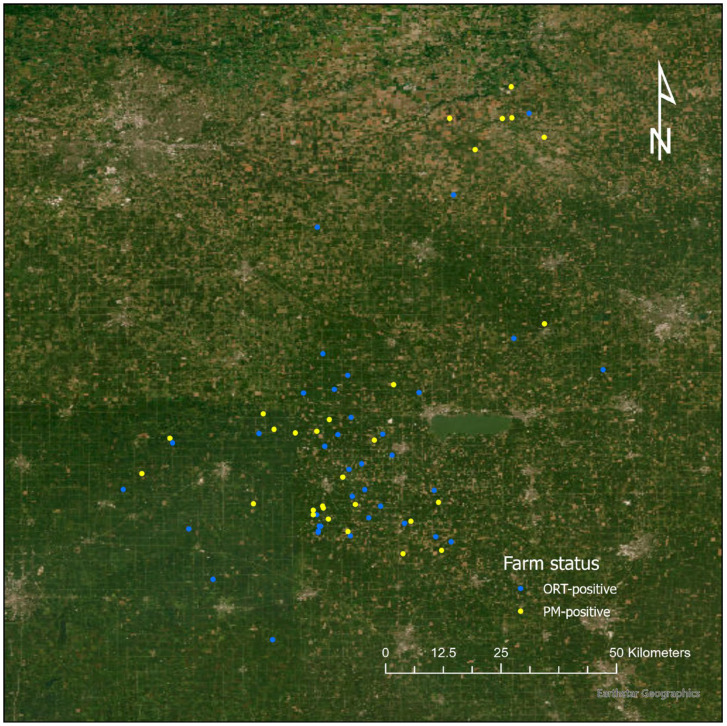
Study region. The geographical references are omitted to mask the exact location of the farms.

**Figure 2 animals-15-00396-f002:**
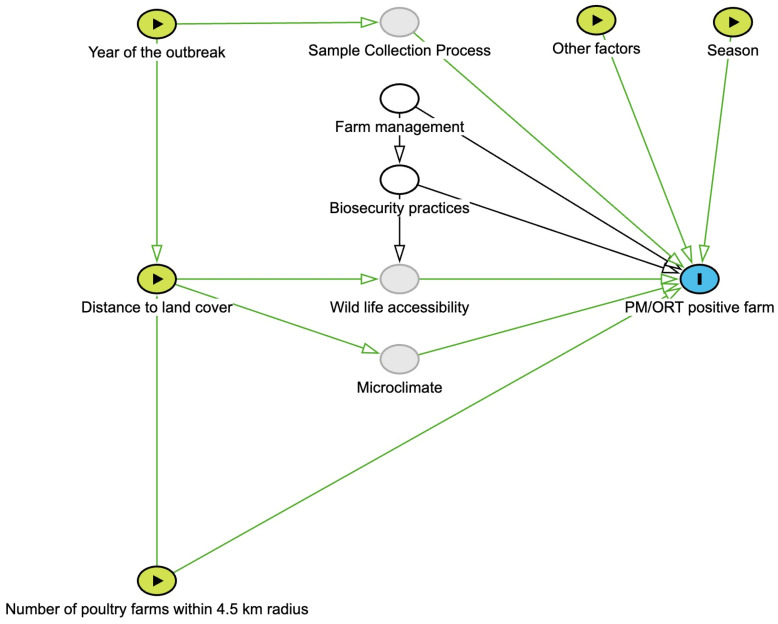
Casual diagram. The green nodes represent risk factors, white nodes are adjusted variables, gray nodes are variables not measured, and the blue node is the outcome. Among all variables, the number of poultry farms within 4.5 km, year of the outbreak, and season of the outbreak were considered as confounders. Farm management and biosecurity practices were adjusted as all farms were from the same production company.

**Table 1 animals-15-00396-t001:** Descriptive statistics and univariable analysis. The outcome of interest was farms with PM compared to ORT. All variables with *p*-value less than 0.2 were selected to next step.

Variables	PM	ORT	Odds Ratio (95% CI)	*p*-Value ^1^
Number of farms	28	37		
# of poultry farms within 4.5 km radius, mean (SD)	9.18 (5.65)	9.32 (5.55)	0.9952 (0.9103, 1.0881)	0.916
Dist. to the nearest wetland (m), mean (SD)	704.28 (458.12)	1338.59 (663.29)	0.9979 (0.9967, 0.9991)	0.001 ***
Dist. to the nearest forest (m), mean (SD)	272.12 (182.58)	412.15 (241.67)	0.9968 (0.9942, 0.9995)	0.018 *
Dist. to the nearest herbaceous land (m), mean (SD)	817.96 (601.09)	861.26 (722.54)	0.9999 (0.9992, 1.0006)	0.794
Dist. to the nearest barren land (m), mean (SD)	2318.34 (1557.53)	3284.17 (1554.09)	0.9996 (0.9993, 0.9999)	0.020 *
Dist. to the nearest water body (m), mean (SD)	1040.63 (798.71)	1282.29 (879.24)	0.9996 (0.9990, 1.0003)	0.260

Distance to the nearest pastureland, N (%) ^2^				0.694
Far	13 (46.43)	19 (51.35)	Ref.	
Near	15 (53.57)	18 (48.65)	0.8210 (0.3071, 2.1953)	
Season, N (%)				0.105
Autumn	8 (28.57)	3 (8.11)	Ref.	
Spring	8 (28.57)	16 (43.24)	0.1875 (0.0388, 0.9058)	
Summer	7 (25)	7 (18.92)	0.3750 (0.0691, 2.0338)	
Winter	5 (17.86)	11 (29.73)	0.1705 (0.0312, 0.9299)	
Year of the outbreak, N (%)				0.001 ***
2014–2017	19 (67.86)	3 (8.11)	Ref.	
2018–2019	3 (10.71)	26 (70.27)	0.0182 (0.0033, 0.1003)	
2020–2021	6 (21.43)	8 (21.62)	0.1184 (0.0236, 0.5945)	

^1^ Continuous variables used Wald Test; categorical variables used Likelihood Ratio Test. ^2^ Categorized in the median (3956.8 m). * *p* < 0.05. *** *p* < 0.001.

**Table 2 animals-15-00396-t002:** Results of the final multivariable logistic regression model for analysis of risk factors for farms with PM isolates compared to ORT.

Variables	OR (95% CI)	*p*-Value (WT) ^1^	*p*-Value (LRT) ^2^
Dist. to the nearest wetland (m)	0.9976 (0.9960, 0.9992)	0.004 **	
Year of the outbreak			0.001 ***
2014–2017	Ref.	Ref.	
2018–2019	0.0148 (0.0020, 0.1082)	0.001 ***	
2020–2021	0.0677 (0.0083, 0.5537)	0.012 **	

^1^ Wald Test. ^2^ Likelihood Ratio Test. ** *p* < 0.05. *** *p* < 0.001.

## Data Availability

De-identified data and non-proprietary data may be accessed by request.

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
