# Peer review of "Exploring the Impact of Land Cover on the Occurrence of Ornithobacteriosis and Fowl Cholera: A Case-Case Study"

_animals, 2025, doi:10.3390/ani15030396_

Round 1
Reviewer 1 Report
Comments and Suggestions for Authors
The aim of this study was to investigate the potential association between land cover type and the incidence of respiratory disease in commercial turkeys. The study focuses on two bacterial respiratory pathogens, Pasteurella multocida (PM) and Ornithobacterium rhinotracheale (ORT), although the title of the study suggests that the manuscript will cover a wider range of respiratory diseases in turkeys. The results indicate that farm location near wetlands significantly increases the incidence of PM and that for every 1-meter increase in distance from a farm to the nearest wetland, the odds of a confirmed disease occurrence of PM decreased by approximately 0.24% compared to an ORT-related disease occurrence.
Although the methodology used in the study is innovative and noteworthy, I do not believe that the results of the study add much important novelty.
The study has several weaknesses/limitations that affect the results and the quality of the manuscript. Some of these have already been mentioned by the authors, i.e. the data on the occurrence of PM and ORT are derived from a single production company with farms located in a relatively narrow geographical area; the data sets used in this study may be subject to unmeasured inaccuracies, particularly with regard to wetlands. This is a very important limitation considering that the main finding of the study is that the farms located closer to wetlands had higher odds of PM compared to ORT. The authors do not present data on farm capacity (number of houses/barns on the farm), modifications or changes in biosecurity measures during the observed period. These may explain why outbreaks of PM decreased significantly in the last two observation periods compared to the first, reference period. It should also be taken into account that ORT can also be transmitted vertically, which is not the case for PM.
Reviewer 2 Report
Comments and Suggestions for Authors
Great manuscript. The introduction, methods, and results are well organized. It is easy to read, and the results and discussion are presented concisely.
My only recommendation/suggestion to the authors would be to add references to the statistical analysis sub-section, if possible. This will strengthen the robustness of the statistical methods utilized for this study.
Author Response
Please see the attachment, thank you.

Reviewer 3 Report
Comments and Suggestions for Authors
Thank you for the opportunity to review this manuscript. I enjoyed reading this work and had only minor feedback for consideration.
- For the reader who is unfamiliar with the Midwestern US, how large of a geographic area were the 65 farms distributed across?
- Were any of the farms located within the same land cover area? i.e., were two or more farms within 4.5km of the same wetland or other region? In reading the methodology and results, I wondered whether there might be any clustering of farms by geographic location, whereby farms within a particular area might be more similar to one another than to those in another area, or conversely, were these sufficiently far apart within the study area that they might have been subject to different weather conditions (i.e., volume and frequency of rainfall), which may have impacted study variables/outcomes.
- Was the fit of the final model assessed?
Overall I was curious whether there might have been any clustering of farms geographically, and if so, whether the authors considered including a random intercept for geographic sub-region (if applicable).
Author Response
Please see the attachment, thank you.

Reviewer 4 Report
Comments and Suggestions for Authors
Peer review:
Manuscript "Exploring the impact of land cover on turkey respiratory dis- 2 eases: a case-case study” (animals-3366344)
Comments to the authors:
The authors conducted a retrospective observational case-case study to investigate the association between land cover and confirmed disease occurrences attributed to Pasteurella multocida (PM) and Ornithobacterium rhinotrachealis (ORT) in commercial turkey sites located in the Midwestern U.S. These bacterial pathogens infect turkeys causing diseases with important losses in the United States commercial turkey industry. In the study, 65 farms from one poultry production company were included, where 28 had PM disease occurrences and 37 had ORT disease occurrences between 2014 and 2021. Some risk factors of interest were different land cover types (wetlands, forest, urban, pasture, herbaceous, barren, shrub), poultry farm density in the area, and season and year of confirmed outbreak(s). A multivariable logistic regression model revealed that for every 1-meter increase in distance from a farm to the nearest wetland, the odds of a confirmed disease occurrence related to PM decreased by approximately 0.24% compared to an ORT-related disease occurrence (P = 0.004). Meanwhile, PM occurrence during 2014-2017 was 98.52% higher than 2018-2019, and 93.23% higher than 2020-2021. The authors state that results indicate that adjacency to wetlands is a significant risk factor for the occurrence of PM disease incidence in commercial turkey farms.
In my opinion, the study is interesting and brings some scientific novelty, despite the fact that the main data were obtained from only one (probably large) production company. The findings are important for the management of two important pathogens on turkey farms.
However, I have one main question for the authors about the study design: why was a case-case-control study not chosen? It would be very important to include the farms negative for both pathogens. The sampling size would be much larger. Since the authors have access to the company's information, they should have the data from the negative farms as well. The overall data (which could be described in section 2.2 of the Materials and Methods, for example) would allow for an overview of the variables of interest across the company's farms. The study must be even better if the authors can compare the risk factors (with statistical analysis) on farms that are negative and positive for the two pathogens.
In addition, I have some suggestions to improve the manuscript:
1) Introduction: since both pathogens are bacteria with similar modes of transmission, I think the introductory text would be improved with a specific paragraph (the 4th one) for transmission and epidemiology. The sentences in the lines 60 to 63 and 70 to 71 should be transferred to this new paragraph. It also should be enriched with more epidemiological information about dissemination of these pathogens. It should be included before the paragraph highlighting the importance of land cover in pathogen dissemination.
2) Materials and Methods: I missed a description of the procedure used by the company to define cases as positive or negative. Was bacterial isolation performed? Serological testing? Molecular testing? How were the analyses performed to detect both pathogens? And were the same methodologies used in all analyses?
3) Discussion: as highlighted in the title, summary and main text, the goal of this study was to assess the importance of land cover as a risk factor for two turkey bacterial respiratory pathogens (PM and ORT). However, the Discussion is mainly focused on wetlands. Expanding the Discussion to address other risk factors analyzed is necessary.
After the preparation of a new version, the manuscript needs to be peer reviewed again. .
Author Response
Please see the attachment, thank you.

Round 2
Reviewer 1 Report
Comments and Suggestions for Authors
The authors provided satisfactory responses to the questions/issues raised in the first review and have made the necessary revisions to the manuscript. I have no further comments.
Author Response
We would like to take the opportunity to thank the reviewer for the comments and feedback.
Reviewer 4 Report
Comments and Suggestions for Authors
Peer review:
Manuscript "Exploring the impact of land cover on turkey respiratory diseases: a case-case study” (animals-3366344R1)
Comments to the authors:
As I previously mentioned in my first review, the authors conducted a retrospective observational case-case study to investigate the association between land cover and confirmed disease occurrences attributed to Pasteurella multocida (PM) and Ornithobacterium rhinotrachealis (ORT) in commercial turkey sites located in the Midwestern U.S. The study is interesting and brings some scientific novelty, despite the fact that the main data were obtained from only one (probably large) production company. The findings are important for the management of these two important pathogens on turkey farms.
In this new version (R1), the authors improved the manuscript according to some of my recommendations. However, two very important modifications were missing, as follows:
1) The authors explained the choice for a case-case study (and not a case-case-control study) in the Methodology. This brief explanation should be transferred to the Discussion. In addition, discuss better this question as answered in the rebuttal letter to me (“The feasibility of a case-case-control approach was considered at the beginning of study. The two main reasons for choosing a case-case study design were data availability and the challenges in defining disease-negative farms in the lack of purposefully collected and tested samples from diagnostics. First, as a retrospective observational study using secondary data, we used the data that was already collected by the production company. The authors did not have access to all the 2 farms of the company, but solely those that had a disease incidence over the observed period of time. During the data collection process over the years, onsite veterinarians assessed disease occurrences and collected samples based on clinical signs. Following collection, samples were cultured and analyzed in-house to identify the pathogen present. Farms with positive samples were confirmed to be disease-positive with certainty. However, for farms without sample collection, we could not definitively confirm they are free of the disease. This lack of certainty prevented us from establishing a reliable pool of “negative” farms for comparison.”)
2) Materials and Methods: I am still missing the description of the procedure used by the company to define cases as positive or negative. Was bacterial isolation performed? Serological testing? Molecular testing? How were the analyses performed to detect both pathogens? And were the same methodologies used in all analyses? Please describe in the Materials and Methods. And not only “After bacterial isolation conducted on farms, the positive samples were sent to the Animal Disease Diagnostic Laboratory (ADDL) under the Ohio Department of Agriculture (ODA), for serological and molecular analyses to confirm specific isolate genotypes.”
After the preparation of a new version, the manuscript needs to be peer reviewed again.
Author Response
Thank you very much for taking the time to review this manuscript. Please find the detailed responses below and the corresponding revisions highlighted in the re-submitted files.
Comment 1: The authors explained the choice for a case-case study (and not a case-case-control study) in the Methodology. This brief explanation should be transferred to the Discussion. In addition, discuss better this question as answered in the rebuttal letter to me (“The feasibility of a case-case-control approach was considered at the beginning of study. The two main reasons for choosing a case-case study design were data availability and the challenges in defining disease-negative farms in the lack of purposefully collected and tested samples from diagnostics. First, as a retrospective observational study using secondary data, we used the data that was already collected by the production company. The authors did not have access to all the 2 farms of the company, but solely those that had a disease incidence over the observed period of time. During the data collection process over the years, onsite veterinarians assessed disease occurrences and collected samples based on clinical signs. Following collection, samples were cultured and analyzed in-house to identify the pathogen present. Farms with positive samples were confirmed to be disease-positive with certainty. However, for farms without sample collection, we could not definitively confirm they are free of the disease. This lack of certainty prevented us from establishing a reliable pool of “negative” farms for comparison.”)
Response 1: Thank you for your comment, the following part has added to the discussion:
“After bacterial isolation conducted by personnel at the production system-level, positive samples were sent to the Animal Disease Diagnostic Laboratory (ADDL) under the Ohio Department of Agriculture (ODA) to be banked and frozen. For this and an accompanying project [33], confirmation of positive farms was done at the ADDL, where samples were cultured, and molecularly characterized as described below. These processes were conducted separately: in brief; for PM, bacterial cultures were conducted using blood plates incubating at 37°C for 48h. Isolate characterization was performed using matrix-assisted laser desorption ionization time-of-flight mass spectrometry (MALDI-TOF MS). Subsequently, PM isolates underwent whole-genome sequencing (WGS) using Il-lumina MiSeq platform with genome libraries prepared via Nextera XT DNA sample prep kit; with this work being conducted at the University of Maryland. For ORT, bacterial culture was performed in blood agar under microaerophilic conditions at 37°C for 48 hours. Isolates were then confirmed using real-time PCR, and WGS was then performed in a consistent manner as described for PM. Further details on the molecular characterization and genotyping methods are available in Campler et al. [33].” (lines 123-138)
Comment 2:  Materials and Methods: I am still missing the description of the procedure used by the company to define cases as positive or negative. Was bacterial isolation performed? Serological testing? Molecular testing? How were the analyses performed to detect both pathogens? And were the same methodologies used in all analyses? Please describe in the Materials and Methods. And not only “After bacterial isolation conducted on farms, the positive samples were sent to the Animal Disease Diagnostic Laboratory (ADDL) under the Ohio Department of Agriculture (ODA), for serological and molecular analyses to confirm specific isolate genotypes.”
Response 2: Thank you for your comment, the following part has added to methodology:
“Another important piece of information to note when interpreting study findings was that, for our study; we did not have data on ‘negative’ or ‘healthy’ farms, and as such we were unable to use a traditional ‘case-control’ study design. The two main reasons for choosing a case-case study design were data availability and the challenges in defining disease-negative farms in the lack of purposefully collected and tested samples from diagnostics. First, as a retrospective observational study using secondary data, we used the data that was already collected by the production company. We did not have access to all the farms of the company, but solely those that had a disease incidence over the ob-served period. During the data collection process over the years, onsite veterinarians assessed disease occurrences and collected samples based on clinical signs. Farms with positive samples were confirmed to be positive using a combination of in-house bacterial culture and diagnostic laboratory molecular testing. However, for farms without sample collection, we could not definitively confirm they are free of the disease. This lack of certainty prevented us from establishing a reliable pool of ‘negative’ farms for comparison.” (lines 335-349)
Round 3
Reviewer 4 Report
Comments and Suggestions for Authors
Accept.